# Mode of Minimally Invasive Surgery Associated with Venous Thromboembolism Incidence in Gynecologic Cancer Patients

**DOI:** 10.3390/curroncol32120655

**Published:** 2025-11-22

**Authors:** Terry Kara, Selphee Tang, Alon D. Altman, Gregg Nelson, Christa Aubrey

**Affiliations:** 1Faculty of Medicine and Dentistry, University of Alberta, Edmonton, AB T6G 1Z2, Canada; kterry@ualberta.ca; 2Department of Obstetrics and Gynecology, Alberta Health Services, University of Calgary, Calgary, AB T2N 1N4, Canada; selphee.tang@albertahealthservices.ca; 3Department of Obstetrics, Gynecology and Reproductive Sciences, Division of Gynecologic Oncology, Faculty of Medicine, University of Manitoba and CancerCare Manitoba Research Institute, Winnipeg, MB R3T 2N2, Canada; aaltman@cancercare.mb.ca; 4Department of Obstetrics and Gynecology, University of Calgary, Calgary, AB T2N 1N4, Canada; gregg.nelson@albertahealthservices.ca; 5Ariadne Labs, Brigham and Women’s Hospital, Harvard T.H. Chan School of Public Health, Boston, MA 02115, USA; 6Department of Obstetrics and Gynecology, Faculty of Medicine and Dentistry, University of Alberta, Edmonton, AB T6G 1Z2, Canada

**Keywords:** minimally invasive surgery, venous thromboembolism, gynecologic oncology

## Abstract

Venous thromboembolism (VTE) following surgery for cancers of the female reproductive tract is a known complication for which there are robust guidelines. However, the incidence of VTE following minimally invasive surgery is known to be low, and there is controversy around who, if any, should have prophylaxis. Our study contributes to the understanding of the low incidence of VTE in this population but highlights an interesting finding that mode of minimally invasive surgery may play a role, contrary to other findings.

## 1. Introduction

Venous thromboembolism (VTE), including deep vein thrombosis (DVT) and pulmonary embolism (PE), is a known major complication in the post-operative period. The rate of VTE following minimally invasive surgery (MIS) for gynecological malignancies is much lower than that reported for open abdominal surgery for gynecologic malignancies. The rate of VTE following MIS was found to be 0.51% in a recent systematic review and meta-analysis in endometrial cancer patients undergoing MIS [1].

Guidelines have remained vague on VTE prophylaxis before and after MIS in gynecologic oncology. VTE risk has traditionally been estimated by Caprini risk score, which looks at risk factors and subsequently categorizes into low-, moderate-, or high-risk groups. Using the Caprini risk score, patients undergoing surgery for gynecological malignancy would score 2 points for malignancy and another 2 points for major surgery > 45 min, which would put >90% into the high-risk Caprini subgroup [2], which is not reflective of the low VTE rate in this patient population.

Instead, it has been advocated to apply modified score for patients with gynecologic oncology undergoing MIS, whereas those with a score of 5–6 and known high-grade histology, known/suspected advanced stage, or lymphadenectomy, or those with a score of ≥7, are offered extended pharmacologic prophylaxis, in addition to preoperative pharmacologic prophylaxis and mechanical prophylaxis [3]. However, even the need for any pharmacologic prophylaxis in the preoperative setting may not be required [1]. Risk stratification for those who may benefit from VTE prophylaxis would be ideal; however, given the low incidence of VTE in this population, this remains elusive.

Our objective was to identify the VTE incidence within 90 days and the clinical factors associated with VTE following MIS for gynecologic malignancy in three centers in Canada.

## 2. Materials and Methods

### 2.1. Study Design and Population

This was a multi-center retrospective cohort study involving patients undergoing MIS for pathologically confirmed gynecologic malignancies between January 2014 and December 2020 in three tertiary referral centers in Canada: Edmonton, Calgary, and Winnipeg. Inclusion criteria are as follows: adults aged 18 years or older, with a pathologically confirmed gynecologic malignancy including cervical, endometrial, ovarian, fallopian tube, and peritoneal cancer, who underwent MIS including laparoscopic, robotic, and combined laparoscopic and vaginal approaches. Exclusion criteria are as follows: conversion from MIS to open, mini-laparotomy, benign pathology, non-gynecologic malignancy, recurrent carcinoma, and other surgeries 90 days before or after MIS for gynecologic malignancy. The study was approved by the Health Research Ethics Board of Alberta Cancer Committee (HREBA.CC-16-0156) and the Research Ethics Board at the University of Manitoba (HS25308(H 2002:009)).

### 2.2. Data Collection and Variables

Cases of gynecologic oncology MIS were identified in Alberta by the Alberta Cancer Registry, with all the laparoscopic cases identified through Health Records, and in Manitoba, all laparoscopic gynecologic oncology cases were identified through Health Records. For cases meeting the study inclusion criteria, electronic charts were reviewed to obtain data on clinical and preoperative characteristics including age, body mass index (BMI), comorbid conditions, neoadjuvant treatment, preoperative bloodwork, and preoperative diagnosis. Operative characteristics including ASA classification, surgical details, estimated blood loss, and intraoperative complications were collected. Postoperative factors collected included the use of thromboprophylaxis in hospital and extended thromboprophylaxis on discharge, length of hospital stay, and postoperative diagnosis. Study data from all three centers were collected on a standardized form and managed using REDCap [4] electronic data capture tool hosted and supported by the University of Calgary.

The primary outcome was defined as incidence of VTE within 90 days post-surgery. Secondary outcomes were incidence of VTE by cancer type and center, and risk factors associated with VTE.

### 2.3. Statistical Analysis

Descriptive statistics (frequency and percent for categorical variables, and median and interquartile range for non-normally distributed continuous variables) were used to summarize preoperative, operative, and postoperative characteristics of the cohort. The primary and secondary outcomes of incidence of VTE per 1000 surgeries were calculated along with the 95% confidence intervals. To identify possible risk factors for VTE, comparisons of characteristics between those with VTE and without VTE were made using the Chi-square or Fisher’s exact test for categorical variables, and the Mann–Whitney U test for continuous variables. We considered a *p*-value of <0.05 statistically significant. Data were analyzed using SAS software, Version 9.4 (SAS Institute Inc., Cary, NC, USA).

## 3. Results

A total of 2006 patients were identified. In Manitoba, 325 patients were identified, with 90 excluded due to screen failures, and in Alberta (Edmonton and Calgary), there were 1916 patients identified, with 130 excluded due to screen failures, leaving 1786 patients who were included in the final analysis.

Patient demographics and operative/postoperative characteristics are shown in Table 1 and Table 2. The median age at the time of surgery was 61 (IQR 16), with a median BMI of 30.3 kg/m^2^ (IQR 10.3). The most common comorbid conditions were hypertension (39.4%), diabetes (14.8%), cardiovascular disease (2.5%), history of DVT (2.1%), history of PE (1.5%), and previous cerebrovascular disease (1.5%). The postoperative diagnosis is shown in Table 3. The majority had uterine cancer (1510 patients, 85.3%), while there were 204 patients (11.5%) with cervical cancer and 40 (2.3%) who had ovarian cancer. Based on the ERGO/ESTRO/ESP endometrial cancer guidelines prognostic risk groups [5], we found that 4/13 (30.8%) of the VTE events occurred in the low-risk group, while 9/13 (69.2%) occurred in the intermediate, high-intermediate, and advanced prognostic groups, although this was not statistically significant (*p* = 0.075). Similarly, among the cervical cancer cases, the three events that occurred in this group were all Stage 1b or higher (*p =* 0.274) (Table 3).

Of the 1786 patients, 15 developed a VTE within 90 days of surgery. This corresponds to a rate of 8.4 per 1000. Nine had a DVT, five a PE, and one patient developed concurrent DVT/PE. Table 4 shows the distribution of the types of VTE across sites in Canada. These events were distributed equally among the three sites across Canada, with no inter-site differences (Calgary 6/579 patients (1.0%), Edmonton 6/964 patients (0.6%), Manitoba 3/243 (1.2%, *p* = 0.438). Two of the 15 VTEs were diagnosed in hospital before discharge, five occurred <30 days postoperatively, and five occurred 30–90 days after surgery. Three of the 15 VTEs did not have a known day but fell within the 90-day cut-off for this study. There were no deaths related to VTE in this study.

In terms of mechanical VTE prophylaxis, 71.5% had pneumatic compression devices/sequential compression devices in the operating room and 65.6% had mechanical prophylaxis post-operatively on the ward. VTE was not significantly associated with use of mechanical thromboprophylaxis; intraoperative mechanical prophylaxis was used in 8/11 VTE cases (72.7%) vs. 1088/1521 (71.5%) cases in the non-VTE group, with *p* > 0.99. Similarly, for postoperative mechanical prophylaxis, 9/13 (69.2%) in the VTE group and 1085/1654 (65.6%) in the non-VTE group, with *p* > 0.99.

Preoperative pharmacological thromboprophylaxis was given in 1371/1743 (78.7%) of patients, mostly in the form of heparin 5000 units subcutaneously, and was not associated with VTE (12/14 (85.7%) vs. 1359/1729 (78.6%), *p* = 0.75). Postoperative pharmacological thromboprophylaxis was given in hospital in 1471/1746 (84.2%) of patients, while 14/1746 (0.8%) of patients were on therapeutic anticoagulation previously, which was restarted postoperatively. The duration of postoperative pharmacological prophylaxis was often only one dose, as 81.4% of the patients had a length of stay in hospital of zero or one day. Of those who developed VTE, 71.4% (10/14) received thromboprophylaxis while in hospital, 7.1% (1/14) were on anticoagulation previously, and 14.3% (2/14) were started on therapeutic anticoagulation in hospital due to VTE. There was missing information on pharmacologic thromboprophylaxis for one of the 15 VTE cases. In the cohort, 5.0% (87/1749) received extended postoperative thromboprophylaxis, and one patient who went on to develop a VTE was on extended thromboprophylaxis (7.1%), while one was on anticoagulation previously, and two were discharged with therapeutic anticoagulation for their VTE diagnosed in hospital. Overall, VTE was not associated with the use of mechanical or pharmacologic thromboprophylaxis (*p* > 0.99 for use of mechanical thromboprophylaxis while in hospital, *p* = 0.256 for use of postoperative pharmacologic thromboprophylaxis while in hospital, and *p* = 0.512 for use of extended postoperative pharmacologic thromboprophylaxis).

The different modes of MIS were found to have differential rates of VTE (Table 4). Robotic surgery was associated with lower rates of VTE (3/882; 0.34%), as compared to laparoscopic (5/370; 1.35%) or combined vaginal/laparoscopic (7/534; 1.31%), *p* = 0.047. In those who developed VTE, pelvic lymphadenectomy was performed in 13 out of 15 patients (86.7%), as opposed to 1070 out of 1771 (60.4%) patients who did not develop VTE (*p* = 0.038). Adjuvant chemotherapy was also significantly associated with VTE, as 40.0% of patients who developed a VTE received adjuvant chemotherapy, compared to only 15.8% of those who did not develop a VTE (*p* = 0.022). There was a non-significant trend towards increased VTE in those with adjuvant radiotherapy (46.7% of those who developed a VTE received adjuvant radiotherapy, and 26.5% of those who did not (*p* = 0.137)).

No statistical difference was found in this cohort when comparing those with VTE to those without VTE with regard to preoperative factors including comorbid conditions, smoking status, BMI, or preoperative lab values (platelet count, hemoglobin, hematocrit, creatinine), or site of disease. There were also no statistical differences found for operative characteristics including ASA classification, intraoperative blood loss >500 mL (although only 2.5% of the cohort had blood loss >500 mL), or operative time, although the operative time was not available for one site (Calgary, corresponding to 605 charts). Postoperatively, length of stay (LOS) in hospital was not statistically associated with VTE, although in four of the cases of VTE the LOS was 3+ days. Stage and histology were also not associated with VTE in this study.

## 4. Discussion

### 4.1. Summary of Main Results

The rate of VTE following MIS for gynecologic malignancy in three tertiary Canadian centers was 15/1786 (0.84%). Robotic surgery, compared to laparoscopic or combined laparoscopic/vaginal surgery, was associated with a statistically significant decrease in the rate of VTE. The only other factors associated with VTE in this study were pelvic lymphadenectomy and adjuvant chemotherapy. There were low event rates, and risk stratification was not possible due to this.

### 4.2. Results in the Context of Published Literature

Despite the consistent result that incidence of VTE after MIS is low in gynecologic oncology, there remains inconsistency in practice regarding extended thromboprophylaxis. The American College of Obstetricians and Gynecologists (ACOG) most recent recommendations are for routine mechanical thromboprophylaxis, with only consideration of pharmacologic thromboprophylaxis in high-risk patients [6], but the American Society of Clinical Oncology (ASCO) recommends extended thromboprophylaxis (4 weeks) in ‘high-risk’ individuals including those with obesity or other risk factors, whether the surgery is performed open or laparoscopically [7]. A recent AAGL White Paper ERAS guideline for MIS in gynecology recommends the use of a modified Caprini score in oncology patients, where a Caprini score > 5 is interpreted in the context of clinical and pathological features that have been shown to correlate with a higher risk of VTE, such as high-grade histology, stage III/IV disease, and lymphadenectomy [3]. Interestingly, the 2012 CHEST guidelines state that for postsurgical patient populations with an overall VTE risk of 0.5–1.5%, even mechanical thromboprophylaxis would not confer enough benefit to justify widespread use, due to the high number needed to treat [8]. In terms of cost-effectiveness, Bell et al. concluded that an overall VTE risk of 4.8% would be required to warrant the cost of extended thromboprophylaxis [9].

Previous research has still been unsuccessful in developing a definition for those at ‘high risk’ of VTE in this patient population, due to the low incidence of VTE and the minimum requirement for a prospective study to have 16,000 patients to develop this type of risk score [10]. Validation of currently described risk scores from prospective trials is required, such as the modified Caprini score or those suggested by Sandadi et al. that look at BMI > 40 kg/m^2^ and operative time ≥ 180 min [11]. A study by Wagner et al. also found an association with a higher Caprini score cut-off (8 or 9), better predicted VTE in gynecologic oncology patients undergoing MIS [12].

Our study once again shows the low incidence of VTE (15/1786, 0.84%), even 90 days after MIS for gynecologic malignancies. Five of these 15 VTE events occurred after the typical 30-day post-operative period used to study VTE incidence in this population in most other studies. The use of extended thromboprophylaxis has been suggested to prevent these VTE events occurring within 30–90 days, but our study showed only a small number of events (5/1786, 0.3%), which raises questions about the utility of extended thromboprophylaxis. Given the low event rate, a predictive score was not possible.

Previous studies have found no association between the mode of MIS, robotic vs. laparoscopic, and the rates of VTE [13,14]. Cusimano et al. completed a meta-analysis looking specifically at obese patients with endometrial cancer and found that the rates of VTE were similarly 0.5% in patients who underwent either a laparoscopic or robotic hysterectomy (5/1015 vs. 2/388 case). The laparoscopic cases were pulled from a total of 14 studies and the robotic cases from five separate studies, but the cut-off time for when the VTEs occurred were not routinely reported [14]. Chan et al. also looked at obese women with endometrial cancer (BMI >/= 40 kg/m^2^) and compared open, laparoscopic, and robotic surgeries (n = 567, 98 and 422, respectively). They found that VTE events did not differ between the three groups (*p* = 0.19) but did not specify the number of VTE events in the patient populations [13]. Wagar et al. found that in a cohort of 806 patients, the mode of laparoscopic surgery also did not differ between MIS modality (five VTE events in total, with n = 410 robotic, n = 264 multiport laparoscopic, and n = 141 single-port laparoscopic) [15]. Our study found robotic surgery was associated with a lower VTE incidence, as only three of the 15 VTE events occurred in patients who underwent robotic surgery (20%), despite the robotic approach being utilized in 49.4% of the total population (*p* = 0.047). All of the robotic surgeries were performed at one site (Edmonton, AB), but when we looked at inter-site differences in VTE incidence, this was not significant. Operative times were not available for the entire cohort but could contribute to this difference seen.

Most studies performed in the past have excluded patients who received adjuvant chemotherapy or radiotherapy postoperatively. We found a statistically significant link between VTE and patients receiving adjuvant chemotherapy, as previously seen [16], along with a higher proportion of patients in the VTE group receiving postoperative radiotherapy, though not statistically significant. Additionally, based on the ERGO/ESTRO/ESP endometrial cancer guidelines [5] prognostic risk groups, we found that VTE events occurred more frequently in the intermediate, high-intermediate, and advanced prognostic groups, although this was not statistically significant (*p* = 0.075). Similar results were shown in the 2017 study by Barber and Clarke-Pearson, who showed that patients with grade 1 or 2 endometrial cancer had a 3.6% incidence of VTE 6 months post-operatively, as opposed to 6.1–9.2% for grade 3 endometrioid or other high-risk histologies [2].

Previous studies have shown that pelvic lymphadenectomy is associated with an increased risk of VTE [17,18], likely owing to the complexity of the surgical procedure as well as the extent of the disease. Kahr et al. compared the risk of VTE after hysterectomy in benign versus malignant disease and evaluated 5513 patients with endometrial cancer and 45,825 with benign disease and found that 1.1% of patients who underwent pelvic lymphadenectomy developed VTE compared to only 0.2% for those who underwent hysterectomy alone (*p* < 0.001) [17]. Latif et al. looked specifically at VTE rates in 15,101 patients with endometrial cancer undergoing hysterectomy with or without lymphadenectomy and found that VTE incidence was increased when a lymphadenectomy was performed (3.8% vs. 2.3%, *p* < 0.0001) [18]. Our study corroborated these findings, as we found pelvic lymphadenectomy to be associated with VTE, as 13 out of 15 patients (86.7%) who developed VTE underwent full lymphadenectomy as opposed to only 60.6% of the total cohort (*p* = 0.038). This increased risk is also utilized in the aforementioned ERAS guidelines for calculating a modified Caprini score.

### 4.3. Strengths and Weaknesses

This was a retrospective study, and therefore all the inherent bias and limitations apply. There was a large proportion of missing data: 216 cases had missing data, and 109 patients were lost to follow-up. Additionally, we were limited to the information that was documented in the medical record, and patients who may have presented elsewhere with a VTE may have been missed. Operative times for cases at one site (Calgary, AB) were not available due to change in record availability over the study period, and this is a major limitation given this could not be used in analysis and this has previously been shown to be a risk factor for VTE. Due to the limited number of events, multivariable analysis could not be performed, limiting the conclusions that can be drawn. Further, patients in this study did receive some form of thromboprophylaxis, and decisions regarding this were provided dependent and therefore subjective. Additionally, only one site utilized robotic surgery, limiting the validity of this conclusion. The strengths of this study include its large sample size, taken from three tertiary sites in Canada, and we used patient-level data rather than a database, which increases the validity of the results. Additionally, this is real-world pragmatic data, in contrast to most VTE studies that include a large proportion of asymptomatic VTEs identified through routine imaging. This study includes only clinically significant VTE results since routine imaging was not performed.

### 4.4. Implications for Practice and Future Research

This study adds to the growing body of research that does not support the use of extended thromboprophylaxis for patients undergoing MIS for gynecological malignancies. Additionally, due to the low VTE event rates, the utility of preoperative or intraoperative VTE prophylaxis could be questioned. Potential exceptions for high-risk patients need to be further studied and characterized. Our study demonstrates the potential role of the mode of MIS surgery in VTE incidence, which should be further studied, and supports previous findings of increased risk of postoperative chemotherapy and lymphadenectomy in the development of VTE.

## 5. Conclusions

This study furthers the evidence that VTE rates after MIS in gynecologic oncology are low. Extended thromboprophylaxis probably has a limited role. Other potential risk factors identified should be further investigated, including the mode of MIS surgery, which cannot be determined from this study due to low event rates and differences in MIS modalities between sites.

## Figures and Tables

**Table 1 curroncol-32-00655-t001:** Preoperative characteristics.

Characteristic	Total (n = 1786)	VTE (n = 15)	No VTE (n = 1771)	*p*-Value *
Patient age at surgery	61 (IQR 16)	69 (IQR 25)	61 (IQR 16)	0.32
	Range 22–90	Range 33–81	Range 22–90	
Comorbid conditions				
Diabetes	265 (14.8%)	3 (20.0%)	262 (14.8%)	0.478
Cerebrovascular disease	26 (1.5%)	0 (0.0%)	26 (1.5%)	>0.999
Cardiovascular disease	45 (2.5%)	2 (13.3%)	43 (2.4%)	0.053
Hypertension	704 (39.4%)	5 (33.3%)	699 (39.5%)	0.628
History of PE	27 (1.5%)	0 (0.0%)	27 (1.5%)	>0.999
History of DVT	38 (2.1%)	1 (6.7%)	37 (2.1%)	0.277
Other	1200 (67.2%)	12 (80.0%)	1188 (67.1%)	0.410
Smoker at time of surgery	189/1723 (11.0%)	1 (6.3%)	188/1708 (11.0%)	>0.999
BMI	30.3 (IQR 10.3)	29.9 (IQR 9.1)	30.3 (IQR 10.3)	0.821
	Range 16.2–68.7	Range 22.6–41.7	Range 16.2–68.7	
	(missing = 20)	(missing = 0)	(missing = 20)	
Neoadjuvant chemotherapy	11/1785 (0.6%)	0 (0.0%)	11/1770 (0.6%)	>0.999
Neoadjuvant radiotherapy	2 (0.1%)	0 (0.0%)	2 (0.1%)	>0.999
Preop platelet level				0.495
<150 × 10^9^/L	25/1707 (1.5%)	0/14 (0.0%)	25/1693 (1.5%)	
150–400 × 10^9^/L	1626/1707 (95.3%)	13/14 (92.9%)	1613/1693 (95.3%)	
>400 × 10^9^/L	56/1707 (3.3%)	1/14 (7.1%)	55/1693 (3.2%)	
Preop hemoglobin				>0.999
<100 g/L	30/1711 (1.8%)	0/14 (0.0%)	30/1697 (1.8%)	
≥100 g/L	1681/1711 (98.2%)	14/14 (100.0%)	1667/1697 (98.2%)	
Preop hematocrit				0.046
<0.36	72/1708 (4.2%)	2/14 (14.3%)	70/1694 (4.1%)	
0.36–0.46	1526/1708 (89.3%)	10/14 (71.4%)	1516/1694 (89.5%)	
>0.46	110/1708 (6.4%)	2/14 (14.3%)	108/1694 (6.4%)	
Preop creatinine				>0.999
<40 umol/L	3/1669 (0.2%)	0/14 (0.0%)	3/1655 (0.2%)	
40–100 umol/L	1586/1669 (95.0%)	14/14 (100.0%)	1572/1655 (95.0%)	
>100 umol/L	80/1669 (4.8%)	0/14 (0.0%)	80/1655 (4.8%)	

**Table 2 curroncol-32-00655-t002:** Operative and postoperative characteristics.

Characteristic	Total (n = 1786)	VTE (n = 15)	No VTE (n = 1771)	*p*-Value *
ASA classification				0.685
1–2	1190/1410 (84.4%)	9/11 (81.8%)	1181/1399 (84.4%)	
3–4	220/1410 (15.6%)	2/11 (18.2%)	218/1399 (15.6%)	
Preoperative pharmacologic thromboprophylaxis	1371/1743 (78.7%)	12/14 (85.7%)	1359/1729 (78.6%)	0.747
Mode of surgery				0.047 *
Robotic	882 (49.4%)	3 (20.0%)	879 (49.6%)	
Laparoscopic	370 (20.7%)	5 (33.3%)	365 (20.6%)	
Combined	534 (29.9%)	7 (46.7%)	527 (29.8%)	
Surgical procedure				
Simple hysterectomy	1595 (89.3%)	12 (80.0%)	1583 (89.4%)	0.211
Bilateral salpingo-oophorectomy	1562 (87.5%)	12 (80.0%)	1550 (87.5%)	0.421
Pelvic lymphadenectomy	1083 (60.6%)	13 (86.7%)	1070 (60.4%)	0.038 *
Pelvic sentinel node biopsy	801 (44.8%)	4 (26.7%)	797 (45.0%)	0.155
Lysis of adhesions	570 (31.9%)	6 (40.0%)	564 (31.8%)	0.579
Others	924 (51.7%)	9 (60.0%)	915 (51.7%)	0.520
Intraoperative blood loss				>0.999
≤500 mL	1575/1615 (97.5%)	13/13 (100.0%)	1562/1602 (97.5%)	
>500 ml	40/1615 (2.5%)	0/13 (0.0%)	40/1602 (2.5%)	
Intraoperative complications	137/1754 (7.8%)	1/14 (7.1%)	136/1740 (7.8%)	>0.999
Mechanical thromboprophylaxis	1096/1532 (71.5%)	8/11 (72.7%)	1088/1521 (71.5%)	>0.999
Operative time (minutes)	166 (IQR 55)	171 (IQR 23)	166 (IQR 55)	0.738
	Range 24–458	Range 135–200	Range 24–458	
	(missing = 605)	(missing = 7)	(missing = 598)	
Mechanical thromboprophylaxis while in hospital	1094/1667 (65.6%)	9/13 (69.2%)	1085/1654 (65.6%)	>0.999
Postoperative pharmacologic thromboprophylaxis in hospital				<0.001
Yes	1471/1746 (84.2%)	10/14 (71.4%)	1461/1732 (84.4%)	
No—pt was on preoperatively	14/1746 (0.8%)	1/14 (7.1%)	13/1732 (0.8%)	
No—none in hospital	259/1746 (14.8%)	1/14 (7.1%)	258/1732 (14.9%)	
No—started due to VTE in hosp	2/1746 (0.1%)	2/14 (14.3%)	0/1732 (0.0%)	
Extended postoperative pharmacologic thromboprophylaxis				<0.001
Yes	87/1749 (5.0%)	1/14 (7.1%)	86/1735 (5.0%)	
No—pt was on preoperatively	55/1749 (3.1%)	1/14 (7.1%)	54/1735 (3.1%)	
No—none upon discharge	1606/1749 (91.8%)	10/14 (71.4%)	1596/1735 (91.9%)	
No—started due VTE in hosp	2/1749 (0.1%)	2/14 (14.3%)	0/1735 (0.0%)	
Length of hospital stay (days)				0.062
0–1	1427/1752 (81.4%)	10/14 (71.4%)	1417/1738 (81.5%)	
2–7	315/1752 (18.0%)	3/14 (21.4%)	312/1738 (18.0%)	
>7	10/1752 (0.6%)	1/14 (7.1%)	9/1738 (0.5%)	

**Table 3 curroncol-32-00655-t003:** Postoperative diagnosis.

Characteristic	Total (n = 1786)	VTE (n = 15)	No VTE (n = 1771)	*p*-Value *
Organ site				0.636
Uterus	1510/1770 (85.3%)	12 (80.0%)	1498/1755 (85.4%)	
Cervix	204/1770 (11.5%)	3 (20.0%)	201/1755 (11.5%)	
Ovary/Fallopian Tube	40/1770 (2.3%)	0 (0.0%)	40/1755 (2.3%)	
Concurrent primary	16/1770 (0.9%)	0 (0.0%)	16/1755 (0.9%)	
Uterine cancer				0.126
Stage 1a, Grade 1 or 2	834/1510 (55.2%)	4/12 (33.3%)	830/1498 (55.4%)	
Other	677/1510 (44.8%)	8/12 (66.7%)	668/1498 (44.6%)	
Cervical cancer				0.272
Stage 1a	83/204 (40.7%)	0/3 (0.0%)	83/201 (41.3%)	
Stage 1b or higher	121/204 (59.3%)	3/3 (100.0%)	118/201 (58.7%)	
FIGO Stage				0.335
1–2	1609/1747 (92.1%)	13 (86.7%)	1596/1732 (92.1%)	
3–4	138/1747 (7.9%)	2 (13.3%)	136/1732 (7.9%)	
Adjuvant chemotherapy	285/1784 (16.0%)	6 (40.0%)	279/1769 (15.8%)	0.022
Adjuvant radiation therapy	476/1784 (26.7%)	7 (46.7%)	469/1769 (26.5%)	0.137

**Table 4 curroncol-32-00655-t004:** VTE within 90 days after surgery.

	Frequency	Rate Per 1000	95% Confidence Interval
Venous thromboembolism	15/1786	8.4	4.2 to 12.6
Type of VTE			
DVT	9/1786	5	1.8 to 8.3
PE	5/1786	2.8	0.3 to 5.2
DVT and PE	1/1786	0.6	0.0 to 3.1
VTE by site *			
Calgary	6/579	10.4	2.1 to 18.6
Edmonton	6/964	6.2	1.3 to 11.2
Manitoba	3/243	12.3	2.6 to 35.7
VTE by mode of surgery **			
Robotic	3/882	3.4	0.7 to 9.9
Laparoscopic	5/370	13.5	4.4 to 31.3
Combined	7/534	13.1	5.3 to 26.8

* *p* = 0.438 for comparing VTE rates by site; ** *p* = 0.047 for comparing VTE rates by mode of surgery.

## Data Availability

Data is stored on an institutional secure REDCap database, complying with local REB. Raw data can be available on reasonable request.

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
