# Peer review of "Curr. Oncol.2025, 32(12), 655;https://doi.org/10.3390/curroncol32120655"

_curroncol, 2025, doi:10.3390/curroncol32120655_

Round 1

Reviewer 1 Report

Comments and Suggestions for Authors

Retrospective study on a relative large cohort of patients. The low incidence of VTE limits the statistical power of the study. Another bias for the study could be the robotic arm - procedures performed only in one site, where the protocol of VTE could be different.

Author Response

Retrospective study on a relative large cohort of patients. The low incidence of VTE limits the statistical power of the study. Another bias for the study could be the robotic arm - procedures performed only in one site, where the protocol of VTE could be different.

Response to reviewers: 

-this has been more explicitly stated in the limitations section as well as the conclusion statement

Reviewer 2 Report

Comments and Suggestions for Authors
  1. General concept comments
  • The abstract contains valuable information, but it could follow the structure of the study for clarity: introduction, methods, results, discussions, and conclusions.
  • The introduction includes relevant information, but it needs more data from the literature regarding the risk factors of VTE and its incidence across different pathologies.
  • At the end of the introduction, the objectives of the study are mentioned, but the aim should be clearly stated in a separate paragraph.
  • The inclusion and exclusion criteria are clearly mentioned.
  • The ”Results” section is difficult to follow. The paragraphs are too long and lose clarity; it deserves to be revised.
  • The tables and figures clearly support understanding the text.
  • The statements in the ”Discussion” chapter are coherent and supported by the cited references.
  • The study's conclusions are important, despite its several limitations. However, these are not explicitly stated in the conclusion section, but must be understood from the text.
  • The cited references are mostly relatively recent publications (from the last 10-15 years) and are appropriate for the discussed topic.
  • The style used for references is not consistent and needs to be revised. Possibly, a few citations could be added for the introduction section and to improve the documentation of the literature.

  1. Specific comments
  • Line 46 – Correct the word ”revie” to "review"
  • Lines 48-51 – You should rephrase the text for clarity: ”VTE risk has traditionally been estimated using the Caprini risk score, which evaluates risk factors and then categorizes patients into low, moderate, or high-risk groups. Patients undergoing surgery for their gynecological malignancy score 2 points for malignancy and 2 points for major surgery lasting over 45 minutes”.
  • Lines 139-141 - Rephrase for better clarity, for example: ”VTE was not significantly linked to the use of mechanical thromboprophylaxis; intraoperative mechanical prophylaxis was used in 8 of 11 VTE cases (72.7%) compared to 1,088 of 1,521 non-VTE ...”
  • Line 167 – Replace ”as opposed to” with ”compared to”, to avoid repetition.
  • Line 170 – Replace ”that” with ”of those who”
  • Lines 170-172 – Rephrase the text for better clarity: ”Adjuvant radiotherapy showed a similar trend but without statistical significance (46.7% of patients with a VTE received adjuvant radiotherapy, compared to only 26.5% of those without one (p=0.137)).”
  • Lines 173-175 – You should rephrase the text as I suggest: ”No statistical difference was found in this cohort when comparing those with VTE to those without VTE regarding preoperative factors, including ......”
  • Line 194 – Replace ”with regards to” with ”regarding”
  • Line 207 – Replace ”to warrant generalized used” with ”to justify widespread use” for clarity and to avoid repetition.
  • Lines 221-224 – Rephrase for clarity, such as: ”The use of extended thromboprophylaxis has been suggested to prevent these VTE events occurring within 30-90 days, but our study showed only a small number of events (5/1786, 0.3%), which raises questions about the utility of extended thromboprophylaxis.”
  • Lines 246-248 – Rephrase the text as I suggest: ”We found a statistically significant link between VTE and patients receiving adjuvant chemotherapy, as previously seen [16], along with a higher proportion of patients in the VTE group receiving postoperative radiotherapy, though this difference was not statistically significant.”
  • Lines 257-258 – Missing words: ”Previous studies have shown that pelvic lymphadenectomy is associated with an increased risk ..”
  • Line 281 – Replace ”on” with ”regarding”
  • Lines 284-287 – You should rephrase this sentence because it loses clarity. ”Additionally, this is real-world pragmatic data, in contrast to most VTE studies that include a large proportion of asymptomatic VTE identified through routine imaging. This study includes only clinically significant VTE results since routine imaging was not performed.”
  • Line 288 - The paragraph with ”Implications for Practice and Future Research” should be mentioned in the ”Conclusion” section.
  • Line 299 – Replace the word ”likely” with ”probably”
Comments on the Quality of English Language

The text deserves a review of the English language and academic style; please follow the suggested recommendations.

Author Response

  1. General concept comments
  • The abstract contains valuable information, but it could follow the structure of the study for clarity: introduction, methods, results, discussions, and conclusions.-
    •  the abstract does follow this format, but it is supposed to be unstructured as per article guidelines
  • The introduction includes relevant information, but it needs more data from the literature regarding the risk factors of VTE and its incidence across different pathologies.
    • We have made the introduction a bit more clear in this regard, but feel that going into risk factors for VTE is not relevant, as there is not enough evidence to elucidate this in the MiS patient population. It is stated the risk score utilized more generally.
  • At the end of the introduction, the objectives of the study are mentioned, but the aim should be clearly stated in a separate paragraph.
    • This has been separated in a different paragraph for clarity
  • The inclusion and exclusion criteria are clearly mentioned.
  • The ”Results” section is difficult to follow. The paragraphs are too long and lose clarity; it deserves to be revised.- done
  • The tables and figures clearly support understanding the text.
  • The statements in the ”Discussion” chapter are coherent and supported by the cited references.
  • The study's conclusions are important, despite its several limitations. However, these are not explicitly stated in the conclusion section, but must be understood from the text.- done
  • The cited references are mostly relatively recent publications (from the last 10-15 years) and are appropriate for the discussed topic.
  • The style used for references is not consistent and needs to be revised. Possibly, a few citations could be added for the introduction section and to improve the documentation of the literature. done

  1. Specific comments
  • Line 46 – Correct the word ”revie” to "review"- done
  • Lines 48-51 – You should rephrase the text for clarity: ”VTE risk has traditionally been estimated using the Caprini risk score, which evaluates risk factors and then categorizes patients into low, moderate, or high-risk groups. Patients undergoing surgery for their gynecological malignancy score 2 points for malignancy and 2 points for major surgery lasting over 45 minutes”. - done
  • Lines 139-141 - Rephrase for better clarity, for example: ”VTE was not significantly linked to the use of mechanical thromboprophylaxis; intraoperative mechanical prophylaxis was used in 8 of 11 VTE cases (72.7%) compared to 1,088 of 1,521 non-VTE ...”- done
  • Line 167 – Replace ”as opposed to” with ”compared to”, to avoid repetition.- done
  • Line 170 – Replace ”that” with ”of those who” - done
  • Lines 170-172 – Rephrase the text for better clarity: ”Adjuvant radiotherapy showed a similar trend but without statistical significance (46.7% of patients with a VTE received adjuvant radiotherapy, compared to only 26.5% of those without one (p=0.137)).”- done
  • Lines 173-175 – You should rephrase the text as I suggest: ”No statistical difference was found in this cohort when comparing those with VTE to those without VTE regarding preoperative factors, including ......”- done
  • Line 194 – Replace ”with regards to” with ”regarding”- done
  • Line 207 – Replace ”to warrant generalized used” with ”to justify widespread use” for clarity and to avoid repetition.- done
  • Lines 221-224 – Rephrase for clarity, such as: ”The use of extended thromboprophylaxis has been suggested to prevent these VTE events occurring within 30-90 days, but our study showed only a small number of events (5/1786, 0.3%), which raises questions about the utility of extended thromboprophylaxis.”- done
  • Lines 246-248 – Rephrase the text as I suggest: ”We found a statistically significant link between VTE and patients receiving adjuvant chemotherapy, as previously seen [16], along with a higher proportion of patients in the VTE group receiving postoperative radiotherapy, though this difference was not statistically significant.”- done
  • Lines 257-258 – Missing words: ”Previous studies have shown that pelvic lymphadenectomy is associated with an increased risk ..” - done
  • Line 281 – Replace ”on” with ”regarding”- done
  • Lines 284-287 – You should rephrase this sentence because it loses clarity. ”Additionally, this is real-world pragmatic data, in contrast to most VTE studies that include a large proportion of asymptomatic VTE identified through routine imaging. This study includes only clinically significant VTE results since routine imaging was not performed.”- done
  • Line 288 - The paragraph with ”Implications for Practice and Future Research” should be mentioned in the ”Conclusion” section.- done
  • Line 299 – Replace the word ”likely” with ”probably”-done